

# Relative impacts of global changes and regional watershed changes on the inorganic carbon balance of the Chesapeake Bay

Pierre St-Laurent[1], Marjorie A. M. Friedrichs[1], Raymond G. Najjar[2], Elizabeth H. Shadwick[3], Hanqin Tian[4], Yuanzhi Yao[4], and Edward G. Stets[5]

[1]Virginia Institute of Marine Science, William & Mary, Gloucester Pt, VA
[2]Department of Meteorology and Atmospheric Science, The Pennsylvania State University, University Park, PA
[3]CSIRO, Marine and Atmospheric Research, Hobart, TAS, AU
[4]School of Forestry and Wildlife, Auburn University, Auburn, AL
[5]U.S. Geological Survey, Mounds View, MN

**Correspondence:** Pierre St-Laurent (pst-laurent@vims.edu)

**Abstract.** The Chesapeake Bay is a large coastal-plain estuary that has experienced considerable anthropogenic change over the past century. At the regional scale, land-use change has doubled the nutrient input from rivers and led to an increase in riverine carbon and alkalinity. The Bay has also experienced global changes, including the rise of atmospheric temperature and $CO_2$. Here we seek to understand the relative impact of these changes on the inorganic carbon balance of the Bay between the early

1900's and the early 2000's. We use a linked land-estuarine-ocean modeling system that includes both inorganic and organic carbon and nitrogen cycling. Sensitivity experiments are performed to isolate the effect of changes in: (1) atmospheric $CO_2$, (2) temperature, (3) riverine nitrogen loading and (4) riverine carbon and alkalinity loading. Specifically, we find that over the past century global changes have increased ingassing by roughly the same amount ($\sim 30$ Gg-C yr$^{-1}$) as has the increased riverine loadings. While the former is due primarily to increases in atmospheric $CO_2$, the latter results from increased net ecosystem

production that enhances ingassing. Interestingly, these increases in ingassing are partially mitigated by increased temperatures and increased riverine carbon and alkalinity inputs, both of which enhance outgassing. Overall, the Bay has evolved over the century to take up more atmospheric $CO_2$ and produce more organic carbon. These results suggest that over the past century, changes in riverine nutrient loads have played an important role in altering coastal carbon budgets, but that ongoing global changes have also substantially affected coastal carbonate chemistry.

## 1 Introduction

The well-documented rise in atmospheric $CO_2$ concentrations is one of the most ubiquitous changes in global biogeochemical cycling over the past century (e.g., Keeling et al., 2003). Although the ocean's biological pump maintains atmospheric $CO_2$ significantly lower than it would otherwise be, the uptake of anthropogenic $CO_2$ by the ocean is governed largely by chemical





and physical processes. These processes include the diffusion of $CO_2$ across the air-sea interface, the dissolution of $CO_2$ and
its dissociation into bicarbonate and carbonate ions, and the transport of anthropogenic dissolved inorganic carbon into the
ocean interior by vertical mixing and subduction. Thus, early estimates of the uptake of anthropogenic $CO_2$ by the ocean did
not explicitly include marine biological processes (Oeschger et al., 1975). However, if biological processes change during the
uptake of anthropogenic $CO_2$ into the ocean, then they can alter that uptake. Such changes could occur in at least three ways:

(1) the $CO_2$ invasion itself, which could influence photosynthesis and calcification (Riebesell et al., 2007, 2000), (2) climate
change, which could influence biogeochemistry via warming and changes in mixing and advection (e.g., Sarmiento et al.,
1998), and (3) the delivery of nutrients and carbon via river runoff and atmospheric deposition (Da et al., 2018; Duce et al.,
2008; Ver et al., 1999; Walsh et al., 1981). Coastal regions, especially estuaries, have unique susceptibility to changes due to
their proximity to anthropogenic nutrient and carbon sources, and therefore may be particularly important for understanding

how biological processes may influence the uptake of anthropogenic $CO_2$ by the ocean.

Different perspectives on the role of the coastal ocean in the uptake of anthropogenic $CO_2$ have been proposed over the past
decades (see Cai, 2011, for a review). For example, Walsh et al. (1981) argued that the input of anthropogenic nitrogen to the
ocean by rivers has stimulated primary production and enhanced the ocean's uptake of atmospheric $CO_2$. On the other hand,
Ver et al. (1999) found that increases in the riverine input of organic matter to the ocean has had a larger and counteracting

effect by stimulating heterotrophy. Such disagreements are to be expected given the great heterogeneity of coastal waters and
the differences in their dominant processes.

Process-based biogeochemical models afford the opportunity to isolate the various ways in which the exchange of $CO_2$
between the atmosphere and coastal waters has changed during the industrial period. Such models represent many of the
important forcing mechanisms, such as the essentially global changes of increasing temperature and atmospheric $CO_2$, as well

as regional shifts in the delivery of freshwater, nutrients, carbon and alkalinity by rivers. Despite the considerable advancement
of estuarine biogeochemical models in recent years (Ganju et al., 2016), the relative impact of these global/regional changes
on carbon cycling in coastal waters is not always clear. Here, we examine these changes and quantify them in the context of
the Chesapeake Bay.

The Chesapeake Bay is a coastal-plain estuary and the largest estuary in the continental United States. Its watershed provides

$\sim 80$ km$^3$ yr$^{-1}$ of freshwater with nearly half of this input coming from one river positioned at the northern end of the Bay
(the Susquehanna River; Figure 1). At its southern end, the Bay is in direct contact with the shelf water of the Mid-Atlantic
Bight (Figure 1). This configuration leads to a meridional gradient of salinity but also of dissolved inorganic carbon (DIC) and
total alkalinity (TA) (e.g., Shen et al., 2019a; Friedman et al., 2020). The gradient is apparent throughout the year, although the
seasonal discharge of the rivers (maximum around March–April) modulates the salinity, DIC and TA (e.g., Shadwick et al.,

2019b) especially in the northern part of the Bay (see the observations in Brodeur et al., 2019).

To better understand the evolution of the Bay over the last century, we perform a process-oriented study based on a numerical
model of the Chesapeake Bay. The study includes a set of numerical experiments quantifying the sensitivity of the inorganic
carbon budget to the global and regional changes described above. The paper is structured as follows. The modeling system and
the numerical experiments are described in the next section. The results from the Control experiment (years 2000–2014) are





then presented, compared to observations, and contrasted with sensitivity experiments representative of the period 1900–1914. Finally, the results of the study are discussed in the context of the existing literature and of the ongoing global and regional changes impacting the Chesapeake Bay region.

## 2 Methods

The study uses a numerical model of the Chesapeake Bay (Feng et al., 2015; Irby et al., 2018; Da et al., 2018, with modifica-
tions described below) and includes a total of six numerical experiments (Table 1). The first experiment (Control experiment) represents contemporary conditions with realistic forcings for a period of 15 years (2000–2014). Then, four sensitivity experiments are used to isolate the effect of specific parameters on the inorganic carbon balance: atmospheric $CO_2$ concentration, temperature, and riverine inputs of nitrogen, carbon and alkalinity. In each of those four experiments, the parameter of interest is modified to represent the period 1900–1914 while keeping all other components of the model the same as in the Control
experiment (Table 1). The last of the six experiments includes the four perturbations at once to evaluate potential synergies. All sensitivity experiments are preceded by a 1 year period during which the model solution adjusts itself to the modification. This adjustment period is not part of the 15 year-long experiments (it precedes them) so that all the experiments represent the Bay in a stationary state (trends $\approx 0$).

### 2.1 Control experiment (2000–2014)

#### 2.1.1 Estuarine model

The numerical experiments are based on an implementation of the Regional Ocean Modeling System (ROMS, Shchepetkin and McWilliams (2005)) for the Chesapeake Bay (ChesROMS-ECB; see Da et al., 2018; Feng et al., 2015). The model domain includes the Bay and a portion of the continental shelf (Figure 1) with a curvilinear discretization on the horizontal (resolution $O(1\,\mathrm{km})$ in the Bay) and 20 topography-following levels on the vertical (Xu and Hood, 2006). The model domain assumes permanent coast-
lines and thus no flooding of land areas.

The ROMS physical kernel is coupled to a biogeochemical module (Estuarine Carbon Biogeochemistry, ECB) at every baroclinic timestep (60 seconds) using a positive-definite advection scheme (Smolarkiewicz and Margolin, 1998). The module represents the nitrogen and carbon cycles of the lower trophic levels (Druon et al., 2010) with additional processes specific to estuarine systems (see Da et al., 2018; Feng et al., 2015). The ECB module includes 17 state variables: nitrate ($NO_3^-$), ammo-
nium ($NH_4^+$), oxygen, inorganic suspended solids (ISS), dissolved inorganic carbon (DIC), total alkalinity (TA), phytoplankton, chlorophyll, zooplankton, small/large nitrogen/carbon detritus, and separate semilabile and refractory dissolved organic carbon/nitrogen components (DOC and DON). Hereafter we refer to the sum of nitrate and ammonium as dissolved inorganic nitrogen (DIN). Note that ECB does not represent the oxidation of hydrogen sulfide (see Cai et al., 2017). The equation for each state variable is documented in the Supplementary Material (Tables S3–S6).





A number of modifications are made to the ECB module described in Da et al. (2018). Specifically, the parameters control-ling the growth and fate of phytoplankton are modified to better represent the observed seasonal cycle of the Bay. First, the ini-tial slope of the photosynthesis-irradiance curve is set to 0.04 (W m$^{-2}$ day)$^{-1}$, similar to the 'spring group' of Cerco and Noel (2004). For $T > 20°C$, the maximum phytoplankton specific growth rate is set to $0.6 \exp(0.078T)$ day$^{-1}$ where $T$ is the water temperature in °C and the coefficient 0.078 °C$^{-1}$ is from Lomas et al. (2002). A constant rate of 2.15 day$^{-1}$ is as-

sumed when $T < 20°C$ (as in Feng et al., 2015; Da et al., 2018) to reflect the observed temperature independence in this range (Lomas et al., 2002). The phytoplankton mortality rate is also decreased to 0.05 day$^{-1}$ and the aggregation rate is increased to 0.008 (mmol-N m$^{-3}$ day)$^{-1}$ to better represent the non-zero phytoplankton concentrations observed at depth during the winter period. Finally, a minimum value of 0.6 m$^{-1}$ is enforced for the coefficient of diffuse attenuation to represent the effect of ISS resuspension in the lower part of the Bay. All these model parameters are documented in the Supplementary Material.

**2.1.2  Atmospheric forcing for the Control experiment (2000–2014)**

The model is forced with the atmospheric forcings (North American Regional Reanalysis, NARR, Mesinger et al. (2006)) described in Da et al. (2018). In addition, we assume that atmospheric $CO_2$ concentrations vary slowly over the period 2000–2014 with a mixing ratio represented by a quadratic polynomial:

$$\text{mixing ratio} = 371.19 + 1.86 (t - t_0) + 0.0125 (t - t_0)^2, \tag{1}$$

where $t$ is the time in years and $t_0 = 2001$. The coefficients are based on a fit to historical global values from the period 1950–2011 assembled by Miller et al. (2014) (see Figure 2). Seasonal variations in atmospheric $CO_2$ concentrations are not considered given our focus on long-term changes.

At the air-sea interface, the model calculates $CO_2$ fluxes using:

$$F = k_w \alpha (pCO_{2a} - pCO_{2w}), \tag{2}$$

where $F$ is the flux in mmol-C m$^{-2}$ day$^{-1}$, $k_w$ is the transfer velocity for $CO_2$ (Wanninkhof, 1992, his Equation 3), $\alpha$ is the $CO_2$ solubility in seawater (mmol-C m$^{-3}$ $\mu$atm$^{-1}$; Weiss, 1974), $pCO_{2a}$ is the atmospheric partial pressure of $CO_2$, and $pCO_{2w}$ is the partial pressure of $CO_2$ at the water surface. Note that $F$ is defined as positive for ingassing; we use this convention because the carbon budget of the Bay is being assessed and all carbon sources are treated as positive. An algorithm adapted from Zeebe and Wolf-Gladrow (2001) is applied to compute $pCO_{2w}$ (as in Fennel et al. (2008)) using modeled surface

temperature, salinity, DIC and TA at each model time-step (60 seconds). The algorithm uses the dissociation constants from Mehrbach et al. (1973) as fitted by Millero (1995).

**2.1.3  Oceanic forcing for the Control experiment (2000–2014)**

Oceanic conditions are prescribed at the model open boundary positioned on the continental shelf of the Mid-Atlantic Bight. Temperature, salinity, oxygen, and dissolved nitrogen (organic and inorganic) are derived using a combination of climatology,

*in situ* (*i.e.*, observational) data and satellite data, as described in Da et al. (2018). For TA and DIC, data from 12 cruises





conducted between 2005 and 2006 in the vicinity of the Bay's mouth (Filippino et al., 2009, 2011) were used to derive the following relationships with salinity ($S$):

$$TA = 25.6\,S + 1222, \qquad N = 98, \qquad R^2 = 0.42, \tag{3}$$

$$DIC = 22.6\,S + 1200, \qquad N = 98, \qquad R^2 = 0.32, \tag{4}$$

where TA is in meq m$^{-3}$, DIC is in mmol-C m$^{-3}$, $N$ is the number of measurements and $R^2$ the coefficient of determination. These relationships are combined with the seasonal climatology used for salinity to prescribe TA and DIC at the model open boundary. Note that the same oceanic conditions are used in the 1900–1914 and 2000–2014 experiments since we are primarily interested in historical changes that occurred inside the Bay. We estimate the historical change in DIC on the continental shelf (*i.e.*, the anthropogenic DIC) to be $\sim +50$ mmol-C m$^{-3}$ (e.g., Zunino et al., 2014), a change that would affect the Bay's bottom

DIC by only a small amount ($\sim +2.8\%$, assuming a salinity of $\sim 25$ psu for the bottom layer of shelf water entering the Bay and using Eq. 4).

### 2.1.4   Riverine forcing for the Control experiment (2000–2014)

At the land/estuary interface, the model is linked to the Dynamic Land Ecosystem Model (DLEM, Yang et al., 2015b, a; Tian et al., 2015) as in Feng et al. (2015). The version of DLEM used here has a resolution of 4 km and provides daily fluxes

for 1900–2015 for the entire watershed of the Bay. For this study these fluxes are aggregated into 10 river sources positioned along the Bay (Figure 1) and include freshwater, NO$_3^-$, NH$_4^+$, DON, DOC, and particulate organic nitrogen/carbon (PON and POC). Riverine fluxes of ISS are provided by the Chesapeake Bay watershed model (Shenk and Linker, 2013).

Riverine fluxes of DIC and TA are calculated from the freshwater discharge of DLEM coupled with our best estimates of riverine concentrations. Numerous studies have shown that riverine TA and DIC exhibit interannual and seasonal variability

(e.g., Raymond and Oh, 2009). However, the observational coverage varies considerably from one river to another and thus these are described individually below (in order of decreasing freshwater discharge).

The Susquehanna River is the river with the most extensive observational record. Two timeseries of TA that together span a period of 58 years (1960–2017) are compared in Figure 3a. The blue timeseries is from Raymond and Oh (2009, site 01540500; see Figure 1) and the green timeseries is derived from the United States Geological Survey data (USGS site 01578310; see

Najjar et al. (2020)). The figure shows actual concentrations with no statistical treatment other than a one year moving average to emphasize long-term changes. Both timeseries suggest a long-term increase of $\sim 9$ meq m$^{-3}$ yr$^{-1}$ between 1960 and 2017 which has been attributed to decreasing acid inputs following the decline in coal mining activity (Raymond and Oh, 2009). Given our focus on long-term changes in the Bay, we use the linear trend of Figure 3a in the 2000–2014 experiment and neglect the remaining year-to-year variations visible in the timeseries. A separate analysis of DIC (which was calculated from TA, pH,

and temperature using PHREEQC (Parkhurst and Appelo, 1999); see Najjar et al. (2020) for details on the analysis) suggests that TA and DIC followed similar trajectories over these decades (Figure 3b); therefore we also assume linearly increasing DIC in the 2000–2014 experiment ($\sim 10$ mmol-C m$^{-3}$ yr$^{-1}$). Finally, a seasonal cycle in TA and DIC concentrations is isolated by applying a one month moving average over the original timeseries and then subtracting the one year moving average of





Figure 3a,b. This seasonal cycle is fitted to a sinusoid with a one year period (Figure 3c) to represent the low concentrations
around March and the high concentrations around September associated with the relative contribution of surface runoff and
groundwater (Najjar et al., 2020). This idealized seasonality is superimposed on the linear trend described (see Table 2) so that
correlations on seasonal timescales between concentrations and freshwater discharge are represented in the model.

The Potomac and James Rivers have the next highest freshwater discharge. As for the Susquehanna River, USGS data
(Najjar et al., 2020) are used to parameterize a seasonal cycle for TA and DIC concentrations (Table 2). We do not include a
long term trend for those rivers as the temporal coverage is more limited than for the Susquehanna. A long-term arithmetic
mean of the TA and DIC concentrations is calculated instead (based on years 1975–2005 for the Potomac and 1975–1995 for
the James) and superimposed on the seasonal cycle (Table 2). Annual mean concentrations are also calculated from the USGS
timeseries for the Patuxent (based on years 1985–1999), Rappahannock (1968–1994) and York Rivers (1990–1998; Table 2)
but no attempt was made at parameterizing their seasonal cycle given their smaller discharge and influence on the Bay. The
concentration for the York River is calculated from its two tributaries (the Mattaponi and Pamunkey Rivers) weighted according
to their mean freshwater discharge.

No timeseries of TA or DIC were available for the remaining four rivers (Elk, Chester, Choptank and Nanticoke Rivers)
and thus the zero salinity intercept (785 meq m$^{-3}$) of an alkalinity-salinity relationship derived for the eastern shore of the
Chesapeake Bay is used (Najjar et al., 2020). This value is assumed constant in time (Table 2). For the DIC concentrations of
the same rivers, no relationship is available and thus we assume that the ratio TA:DIC is 1:1 at a salinity of zero (e.g., Figure 2
of Friedman et al., 2020).

With these assumptions, the Bay's mean riverine loading over the period 2000–2014 is 89 Geq yr$^{-1}$ for TA and 1169 Gg-
C yr$^{-1}$ for DIC (Table 3). The Susquehanna River itself contributes to 45% of TA, 45% of DIC, and 47% of freshwater
discharge during this period.

**2.1.5   Data available for the evaluation of the Control experiment**

The hydrodynamic and nitrogen components of the model have been extensively evaluated in previous publications (e.g.,
Irby et al., 2016; Da et al., 2018) based on the data from the monitoring program of the Chesapeake Bay (USEPA, 2012). In or-
der to evaluate the inorganic carbon system of our Control experiment, a dataset of TA, DIC and $p$CO$_{2w}$ is used (Shadwick et al.,
2019a; Friedman et al., 2020). The data were collected along the main stem of the Bay (37°N to 39.5°N) over the period June
2016 to June 2018 and cover all four seasons. The dataset includes a total of 204 data points of surface TA, DIC and $p$CO$_{2w}$.
In order to compare these data with the Control experiment (years 2000–2014), a seasonal climatology was assembled for the
months of December to February, March to May, June to August, and September to November. This combination is chosen so
that all seasons include a comparable number of data points.





## 2.2 Sensitivity experiments

### 2.2.1 Sensitivity to atmospheric CO$_2$ concentration (1900$_{CO2}$)

This sensitivity experiment is designed to isolate the impact of atmospheric CO$_2$ from all other drivers of change. It is identical to the Control experiment except that it uses atmospheric CO$_2$ concentrations that are representative of the early 1900's. More precisely, the experiment assumes a constant mixing ratio of 300 ppm (representative of the values during 1900–1914 in Miller et al. (2014)) which is $\sim$ 90 ppm lower than during 2000–2014 (Figure 2). Note that a change in atmospheric CO$_2$ alone does not affect the primary production and respiration of the model but it does affect the DIC and the carbon budget.

### 2.2.2 Sensitivity to temperature (1900$_T$)

This sensitivity experiment isolates the impact of temperature change by using water temperatures that are always 1.5°C lower than the Control experiment throughout the water column. Few long-term records of water temperature date back to the early 1900's and thus the 1.5°C value should be viewed as an approximation. The 1.5°C corresponds approximately to the difference in water temperature between years 1990–2005 ($\sim$ 16°C) and years 1940–1950 ($\sim$ 14.5°C) at the mouth of the Patuxent River (Najjar et al., 2010, their Figure 3). The uniformity of the change in the vertical is justified by the shallow depths of the Bay and supported by the study of Preston (2004). Note also that in this experiment the change in temperature only affects the biogeochemical fields; the uniform change considered here would have only a minor impact on the physical circulation of an estuary like the Chesapeake Bay, and thus we simply use the same physical fields as in the Control experiment.

With the temperature-dependent formulations used in the model, the historical increase of 1.5°C represents a $\sim$ +11% increase in the maximum phytoplankton growth rate, maximum grazing rate, and remineralization rate. Note that phytoplankton production is also limited by nutrients and light and these two can mitigate the increase expected from temperature alone. Respiration depends on the amount of organic matter present in the water column and is thus expected to mirror changes in production.

### 2.2.3 Sensitivity to riverine inputs of nitrogen (1900$_N$)

This sensitivity experiment isolates the impact of change in nitrogen loading by using riverine concentrations of nitrogen that are modified to represent conditions of the early 1900's. DLEM riverine concentrations of NO$_3^-$, NH$_4^+$, PON and DON for the period 1900–1914 are used for this purpose, following the same protocol as for the Control experiment. However, the river freshwater discharge remains the same as in the Control experiment (2000–2014) so that the physical fields of the model (notably the currents and stratification) are unaffected and thus the differences between the two runs are solely due to the riverine concentrations of nitrogen. Note that the mean freshwater discharge of the Bay's watershed is comparable during the periods 1900–1914 and 2000–2014 (78 and 86 km$^3$ yr$^{-1}$, respectively (DLEM)).

The DLEM results suggest that the main change that occurred between 1900–1914 and 2000–2014 is a large increase in riverine nitrate concentrations (Figure 4) that occurred primarily in the late 1960's and early 1970's. This increase is well





documented (see Harding et al. (2016) and the observations in Figure 4) and is generally attributed to a large increase in nitrogen fertilizer usage after World War II. The Bay's DIN loading increased by 33% between the 1900's and the 2000's (DLEM, Table 3).

### 2.2.4 Sensitivity to riverine inputs of carbon and alkalinity ($1900_C$)

This sensitivity experiment uses riverine fluxes of carbon and alkalinity that are modified to represent conditions of 1900–1914.

The modifications to TA and DIC are limited to the Susquehanna River as it accounts for approximately half of the freshwater discharge to the Bay and it is the only river with $> 50$ years of observations (Figure 3). For simplicity, we assume a constant annually-averaged TA and DIC throughout the period 1900–1914 that is equal to the estimates for year 1960 from the linear trend (Table 2). The idealized seasonal cycle of TA and DIC remains the same as in the Control experiment.

With these assumptions, the Bay's riverine loading increases by 25% (TA) and 22% (DIC) between the early 1900's and

early 2000's (Table 3). Note that such increases in DIC and TA inputs, taken alone, would not affect the primary production and respiration of the model, but they certainly affect the model budgets of inorganic carbon. In the case of total organic carbon (TOC), the Bay's riverine loading increases by 11% from the 1900's to the 2000's (DLEM, Table 3).

### 2.2.5 Combined effect of atmospheric $CO_2$, temperature, riverine N, C and TA ($1900_{all}$)

This numerical experiment simultaneously applies all four perturbations described above (atmospheric $CO_2$, temperature, and

riverine inputs of nitrogen, carbon and alkalinity), allowing us to test the additivity of the changes caused by the individual perturbations. The sum of the four individual perturbation experiments is unlikely to match experiment $1900_{all}$ exactly since the perturbations are not acting independently. As one example, the air-sea $CO_2$ flux ($F$) depends non-linearly on $T$, DIC, and TA through $pCO_{2w}$.

### 2.3 Carbon budgets assembled from the model results

A carbon budget for the Chesapeake Bay (including the tributaries and integrated from surface to bottom) is calculated at every model timestep and then averaged over the simulation periods (15 years). The equations of the budget are (e.g., Wakelin et al., 2012):

$$\frac{\partial}{\partial t} \iiint \mathrm{DIC}\, dV = River\, DIC - Export\, DIC - NEP + Airseaflux, \tag{5}$$

$$\frac{\partial}{\partial t} \iiint \mathrm{TOC}\, dV = River\, TOC - Export\, TOC + NEP - Burial, \tag{6}$$

$$NEP = Production - Respiration, \tag{7}$$

where the terms on the left hand side of Equations 5–6 represent changes in DIC and TOC inventory over time. The first two terms on the right hand side of Equations 5–6 represent the input from rivers and the net horizontal flux (export) across the Bay's mouth (positive seaward). NEP is the net ecosystem production and represents the difference between production and respiration (Equation 7). $Airseaflux$ is the net air-sea $CO_2$ flux over the domain and it is defined positive if this term represents





a source of carbon to the Bay (net ingassing). Burial represents the fraction of the bottom TOC flux that is permanently buried (*i.e.*, not resuspended nor respired).

In the model, the inorganic and organic budgets (Equations 5–6) are closed and there is no residual term. We report budget values rounded to the nearest integer (in Gg-C yr$^{-1}$) as this corresponds to the order of magnitude of the smallest term in the time-averaged budget. Note, however, that the terms have year-to-year variations that far exceed 1 Gg-C yr$^{-1}$. This variance is

quantified by the standard deviation of the annually-averaged budget terms and is indicated with the symbol $\pm$. The standard deviation is rounded to the nearest 10 Gg-C yr$^{-1}$ in the text and in the tables.

## 3  Results

### 3.1  Control experiment (2000–2014)

#### 3.1.1  Overview of the inorganic carbon system

The inorganic carbon system of the model exhibits important **regional differences** (Figure 5). The tributaries of the Bay, including the major Susquehanna River in the north (Figure 1), are associated with relatively low DIC and TA, and produce a gradient increasing seaward along the main stem of the Bay (Figure 5a,b; see also Brodeur et al. (2019) and Friedman et al. (2020)). The lower DIC and TA concentrations are particularly apparent in the northern half of the Bay (Figure 5a,b) where the Susquehanna River delivers $\sim 45\%$ of the freshwater discharge to the Bay. One exception to the low tributary DIC and TA

concentrations is the Potomac River where concentrations are higher than all other rivers (Table 2) and approach those of shelf water in the fall season.

The low TA of the river water is accompanied by relatively high surface $p\mathrm{CO}_{2\mathrm{w}}$ inside the tributaries and downstream of the Susquehanna River (Figure 5c). Away from the tributaries, surface $p\mathrm{CO}_{2\mathrm{w}}$ values are generally close to atmospheric levels ($\sim 385\,\mu$atm in 2000–2014). This spatial distribution of surface $p\mathrm{CO}_{2\mathrm{w}}$ drives strong outgassing within the tributaries and in

the northern half of the Bay, and either ingassing or near-neutral conditions in the southern half (Figure 5d).

The inorganic carbon system also exhibits large **seasonal variations**. The signature of seasonal river inputs is most apparent by comparing the period March–May (after the spring freshet results in low surface salinity), to the period Sep.–Nov. (after the low freshwater inputs of the summer season). In March–May, the fresh riverine water contributes to low DIC and TA concentrations in the estuary whereas Sep.–Nov. shows higher DIC and TA (Figure 5a,b). Similarly, $p\mathrm{CO}_{2\mathrm{w}}$ oscillates seasonally

with lower values in March–May and highest values in Sep.–Nov. (Figure 5c). Biological production and temperature contribute to this seasonality of $p\mathrm{CO}_{2\mathrm{w}}$, with warming temperatures increasing $p\mathrm{CO}_{2\mathrm{w}}$ and increasing production mitigating this (e.g., Friedman et al., 2020). The seasonality of $p\mathrm{CO}_{2\mathrm{w}}$ is reflected in the air-sea $\mathrm{CO}_2$ fluxes and results in strong outgassing or near neutral conditions in Sep.–Nov., while March–May is characterized by weaker outgassing (in the northern Bay) and even strong ingassing in the southern Bay (Figure 5d). In the period June–Aug., $p\mathrm{CO}_{2\mathrm{w}}$ is relatively low (close to atmospheric

concentrations), resulting in near neutral air-sea fluxes (Figure 5c,d).





### 3.1.2 Evaluation of the modeled inorganic carbon system

As noted earlier, the hydrodynamics and nitrogen cycle of the linked DLEM-ChesROMS-ECB modeling system have been well evaluated (Feng et al., 2015; Irby et al., 2016, see also the Supplementary Material for an assessment of the main model variables (Table S1, Figures S1–S2)). Therefore, here the model evaluation is focused on the carbon cycling component of

the model (Figure 5). Regional and seasonal variabilities highlighted in the previous section are generally supported by the observations of Friedman et al. (2020) (see also Brodeur et al., 2019). These include the north-south gradient in properties, the seasonally varying influence of the rivers, and the seasonality of $pCO_{2w}$ away from the tributaries. Some discrepancies are apparent, however. At the mouth of the Bay, the model generally underestimates surface DIC concentrations (Figure 5a). This bias is particularly apparent in March–May and extends into the southern half of the Bay.

The bias in DIC directly affects surface $pCO_{2w}$ which is similarly biased low in the southern Bay in March–May (Figure 5c). We note, however, that this bias is less and less apparent away from the Bay's mouth, and that the modeled $pCO_{2w}$ in the mainstem Bay agrees well with the data points in the vicinity of the Potomac River (Figure 5c). In the northern half of the Bay, observed $pCO_{2w}$ sometimes exhibit noisy patterns (particularly in Sep.–Nov.; Figure 5c) that are not reproduced by the model. The potential causes of these differences will be discussed later (see Section Discussion).

The spatio-temporal variability of the inorganic carbon system can be strongly influenced by biological production within the Bay. This important component of the model is evaluated for the period 2002–2011 using results from an empirical satellite productivity model calibrated with *in situ* observations (Son et al., 2014). The empirical model provides a seasonal climatology of net primary production (NPP) for three subregions of the Bay's main stem (Figure 6). Results from both models exhibit a strong seasonal cycle with peak NPP between the months of May and July (consistent with *in situ* data in Figure 4 of

Harding et al. (2002)) and a similar magnitude of NPP during the summer. They also agree on the differences between the three regions with summer NPP being highest in the upper Bay and lowest in the lower Bay. The primary difference between the two sets of model results is that ChesROMS-ECB generates higher production in the winter months in the lower Bay (Figure 6).

### 3.1.3 Combined inorganic and organic carbon budget

A carbon budget (Equations 5–6) for the Chesapeake Bay domain (including the tributaries but excluding the continental shelf) is calculated over the simulation period of the Control experiment (years 2000–2014, Table 4). If we first consider the total carbon (the sum of inorganic and organic carbon), the budget shows a near balance between riverine carbon inputs and advective output at the mouth of the Bay ('Export'; see Table 4). The difference between the two (196 Gg-C yr$^{-1}$) is equivalent to $\sim 12\%$ of the annual riverine carbon input, and is largely balanced by burial within the Bay ($221 \pm 20$ Gg-C yr$^{-1}$). In comparison with

these terms, the carbon inventory of the Bay shows a very small trend over the 15 years of the simulation but substantial year-to-year variability ($+8 \pm 60$ Gg-C yr$^{-1}$, Table 4).

Despite the near balance between the riverine carbon input and the carbon export at the Bay's mouth, considerable biogeo-chemical transformations take place within the budget domain. Production and respiration are each equivalent to $\sim 270\%$ of





the annual riverine carbon input (Table 4). The difference between production and respiration (NEP) is, however, an order of

magnitude smaller ($+259 \pm 60$ Gg-C yr$^{-1}$, positive indicating net autotrophy; Figure 7b). NEP is thus equivalent to $\sim 15\%$ of the riverine carbon input and is comparable in magnitude to burial (Table 4). The standard deviation of NEP is relatively small ($< 20\%$ of the standard deviation associated with production or respiration) as years of high production are also years of high respiration (not shown).

As noted in Figure 5, the air-sea $CO_2$ flux exhibits ingassing or near-neutral fluxes in the southern half of the Bay, and strong

outgassing within the tributaries and in the northern half of the Bay (*i.e.*, downstream of the Susquehanna River). The net air-sea $CO_2$ flux of the Bay, defined as ingassing minus outgassing, is very close to zero ($+34$ Gg-C yr$^{-1}$, *i.e.* slightly ingassing; Table 4 and Figure 7b). The sign of the net flux is thus sensitive to environmental changes and fluctuates substantially from one year to another (negative during 4 years and positive during 11 years). The standard deviation over 2000–2014 of the net air-sea flux is $\pm 90$ Gg-C yr$^{-1}$.

### 315 3.2 Sensitivity experiment results: Changes in the carbon budget

#### 3.2.1 Experiment 1900$_{CO2}$ versus Control experiment

The increase in atmospheric $CO_2$ concentrations (experiment 1900$_{CO2}$ versus Control experiment) only affects the inorganic component of the carbon budget (Table 4, Figure 7c). The production, respiration, burial and export of organic carbon in experiment 1900$_{CO2}$ are thus identical to the Control experiment. The historical change in atmospheric $CO_2$ is large enough

to reverse the sign of the net air-sea flux from $-20 \pm 90$ Gg-C yr$^{-1}$ (slightly outgassing in 1900$_{CO2}$) to $+34 \pm 90$ Gg-C yr$^{-1}$ (slightly ingassing in the Control experiment; Table 4). The increase in the net air-sea $CO_2$ flux is accompanied by an increase in DIC export of similar magnitude ($+51$ Gg-C yr$^{-1}$, Table 4, Figure 7c). Note that this increase in export reflects higher DIC concentrations within the Bay and not a change in the physical circulation of the Bay. As in the Control experiment, the trends in inorganic and organic carbon are very small over the 15 years of the experiment (Table 4).

#### 325 3.2.2 Experiment 1900$_T$ versus Control experiment

The increase in water temperature (experiment 1900$_T$ versus Control experiment) mostly affects the production and respiration, with increases of $+252$ Gg-C yr$^{-1}$ and $+265$ Gg-C yr$^{-1}$, respectively (Table 4) and only a small resulting change in NEP ($-13$ Gg-C yr$^{-1}$; Figure 7d). The net air-sea $CO_2$ flux over the domain changes from $+57$ Gg-C yr$^{-1}$ to $+34$ Gg-C yr$^{-1}$, *i.e.* the change in temperature brings the Bay closer to being neutral (Figure 7d). Note that the increase in temperature affects

surface $pCO_{2w}$ and contributes to the change in air-sea $CO_2$ flux and to a 13 Gg-C yr$^{-1}$ reduction in the DIC export (Figure 7d, Table 4). The other components of the budget (burial and export of organic carbon) are mostly unchanged by the warming.

#### 3.2.3 Experiment 1900$_N$ versus Control experiment

The increase in riverine inputs of nitrogen (experiment 1900$_N$ versus Control experiment) has a strong impact on production and respiration (Table 4). These two terms are increased by $+492$ Gg-C yr$^{-1}$ and $+391$ Gg-C yr$^{-1}$ (respectively) resulting in a NEP





increase of +101 Gg-C yr$^{-1}$ (Figure 7e, Table 4). This change in NEP affects the organic component of the budget, increasing both burial and TOC export at the Bay's mouth by similar amounts (+55 Gg-C yr$^{-1}$ and +46 Gg-C yr$^{-1}$, respectively). The net air-sea $CO_2$ flux shows the largest change of all the sensitivity experiments, changing from $-36$ Gg-C yr$^{-1}$ (slightly outgassing) to $+34$ Gg-C yr$^{-1}$ (slightly ingassing). This change is consistent with the increase in NEP and with a decrease in surface DIC concentrations. Finally, the increase in riverine inputs of nitrogen produces a decrease in DIC export of 31 Gg-

C yr$^{-1}$ (Figure 7e, Table 4).

### 3.2.4   Experiment 1900$_C$ versus Control experiment

The increase in riverine inputs of carbon and alkalinity (experiment 1900$_C$ versus Control experiment) leads to an increase in the respiration term (+30 Gg-C yr$^{-1}$, Table 4). The latter is solely a result of the increased riverine TOC loading as TA and DIC alone would not affect respiration. Since the production is nearly unchanged, overall the NEP exhibits a decrease of 28 Gg-

C yr$^{-1}$ (*i.e.*, the Bay is becoming less autotrophic). The net air-sea $CO_2$ flux decreases from +76 Gg-C yr$^{-1}$ to +34 Gg-C yr$^{-1}$, meaning that the change in the riverine carbon and TA brings the Bay closer to being neutral (Table 4, Figure 7f). The change in TA and DIC contributes to this change in air-sea $CO_2$ flux by their impact on surface $pCO_{2w}$. Assuming an annually averaged water temperature of 15°C, conservative mixing between the properties of the Susquehanna River (Table 2) and an oceanic end-member defined by $S = 33$ psu and Equations 3–4, we estimate an increase of up to $\sim 9\%$ in surface $pCO_{2w}$ between

the 1900's and the 2000's. Finally, the increase in riverine inputs of DIC is accompanied by a similar increase in DIC export, leading to a small net effect on the horizontal DIC transport (+19 Gg-C yr$^{-1}$; Table 4, Figure 7f).

### 3.2.5   Experiment 1900$_{all}$ versus Control experiment

When all four changes are made simultaneously (*i.e.* experiment 1900$_{all}$ with simultaneously increased atmospheric $CO_2$, temperature, nitrogen loading, DIC and TA loadings; Figure 7a), the results differ from the Control experiment primarily in

terms of air-sea $CO_2$ flux and NEP. The air-sea $CO_2$ flux switches from a small net source to the atmosphere ($-8$ Gg-C yr$^{-1}$) to a net sink (+34 Gg-C yr$^{-1}$), and NEP becomes increasingly autotrophic (209 Gg-C yr$^{-1}$ to 259 Gg-C yr$^{-1}$; Figure 7a). The results generated in Experiment 1900$_{all}$ are also similar to what is obtained by adding the results of the four sensitivity experiments described above (*i.e.* compare the last two columns in Table 4), suggesting a substantial linearity between these four experiments. Some differences do exist, however. Specifically, the changes in the net air-sea $CO_2$ flux, burial and NEP are

all slightly smaller when the four experiments are run simultaneously, than when the results of the four experiments are simply added together.

### 3.3   Relative importance of global and regional changes

The relative importance of global and regional changes is assessed by combining the experiments as follows: 1900$_{CO2}$+ 1900$_T$ (global), and 1900$_N$ + 1900$_C$ (regional). An important result is that this grouping narrows the gap between the early 1900's and

early 2000's by combining changes of opposite signs (Figure 7c–f). For example, the change in air-sea $CO_2$ flux from rising





atmospheric $CO_2$ concentrations is partially mitigated by the increased outgassing from rising temperatures. The net effect of global changes is to decrease the net horizontal flux of DIC ('Rivers − Export') by 38 Gg-C $yr^{-1}$, increase the net air-sea $CO_2$ flux by 31 Gg-C $yr^{-1}$, and decrease the NEP by 12 Gg-C $yr^{-1}$ (Figure 7, Table 4).

In the case of regional drivers, the large increases in ingassing and NEP from increased DIN loadings are partially mitigated by the changes in riverine inputs of carbon and alkalinity (Figure 7e–f). The net effect of regional changes is to increase the net horizontal flux of DIC ('Rivers − Export') by 50 Gg-C $yr^{-1}$, increase the net air-sea $CO_2$ flux by 27 Gg-C $yr^{-1}$, and increase the NEP by 73 Gg-C $yr^{-1}$ (Table 4). Global and regional drivers are thus of similar importance when assessing changes in the inorganic carbon balance, with the exception of NEP which is primarily affected by regional drivers. Note that global and regional drivers both push the Bay toward net ingassing but they influence the horizontal DIC flux and NEP in opposite ways.

## 4   Discussion

### 4.1   Uncertainties and comparison with other studies

The inorganic carbon budget of the Control experiment (Table 4) can be compared to that of Shen et al. (2019b). The two model-derived budgets share the same key features, namely, a positive net horizontal flux ('Rivers − Export') of DIC (234 Gg-C $yr^{-1}$ and 157 Gg-C $yr^{-1}$, respectively), a positive NEP (259 Gg-C $yr^{-1}$ and 165 Gg-C $yr^{-1}$, respectively) and a compara-

tively small net air-sea $CO_2$ flux (34 Gg-C $yr^{-1}$ and $-50$ Gg-C $yr^{-1}$, respectively). The main discrepancy is the riverine DIC loading (1169 Gg-C $yr^{-1}$ in this study and 821 Gg-C $yr^{-1}$ in Shen et al. (2019b)). The cause of this difference is unclear as the two studies assume similar DIC concentrations for the largest river (Susquehanna River; not shown). Potential explanations include differences in the years examined (2000–2014 versus 1986–2015), differences in the riverine freshwater discharge used in the simulations, or differences in the DIC concentrations assumed for the smaller rivers. Because most of the DIC loading

from the rivers is exported to the coastal ocean, these differences are unlikely to cause major discrepancies in the 1900's versus 2000's changes reported in this study.

    The budget of the Control experiment (Table 4) can also be compared to that of Kemp et al. (1997). They estimate riverine loadings of 55.8 Gg-N $yr^{-1}$ (DIN), 39.5 Gg-N $yr^{-1}$ (TON) and 261.8 Gg-C $yr^{-1}$ (TOC). While their TON loading is similar to that in our budget, their DIN and TOC loadings are 42% and 48% lower (respectively) than in our budget. The difference in

DIN loading must originate from the smaller tributaries of the Bay since the DIN loading of the Susquehanna is nearly identical between the two studies. In the case of the TOC loading, the discrepancy remains whether we focus on the Susquehanna or the watershed. The TOC export is also 48% lower in Kemp et al. (1997) than in the present study (with the caveat that the two budgets represent different years). The other components of the budget in Kemp et al. (1997) are not directly comparable to our study as they are specific to the main stem of the Bay.

Although in general the model results represent recent climate-quality data in the Chesapeake Bay quite well (Figures 5) it is worth discussing the origin and impact of the model biases and how these may be reduced in future work. For example, the low DIC bias at the mouth of the Bay (Figure 5a) most likely originates from uncertainties in the DIC concentrations prescribed at the model's oceanic boundary, which are derived from limited measurements (Section Methods). The low DIC bias leads,





in turn, to a low bias in $pCO_{2w}$ in March–August in the southern half of the Bay (Figure 5c). The observed $pCO_{2w}$ values

suggest a relatively weak outgassing in this region and time of the year, while the model exhibits a weak ingassing (Figure 5d). The bias is, however, unlikely to have a major impact on net air-sea $CO_2$ flux of the model as it appears to be geographically confined to the southern part of the Bay (Figure 5c). In future implementations of the model, more climate-quality data will be used to improve this outer boundary condition issue.

Differences in modeled and observed $pCO_{2w}$ are also apparent in the northern half of the Bay (Figure 5c). A possible

explanation for these differences is a temporal mismatch between the observations and the model results (which are from different years; see Methods). Such a mismatch in years can cause substantial differences in the water properties of this area as the freshwater discharge of the Susquehanna River varies substantially between years and often controls the along-shore gradients (Zhang et al., 2006).

Historical changes that were not considered in the present study include alkalinity sinks within tributaries such as the Po-

tomac River (Najjar et al., 2020) due to biogeochemical processes not accounted here. Other historical changes not considered in the present study include the warming, salification and acidification of continental shelf waters (Wallace et al., 2019; Saba et al., 2015). Future studies should consider the role that these oceanic changes have played over the past century. Finally, it is worth pointing out that the important topic of coastal acidification (Cai et al., 2011, 2017) was not examined in this study but that it should be a focal point of future studies.

### 4.2   Changes in Chesapeake Bay carbonate chemistry over the past century

There have been considerable changes to the inorganic carbonate system of the Chesapeake Bay over the past century. Causes include both global factors, including increases in atmospheric $CO_2$ and increases in temperature, as well as more regional factors within the watershed, including increases in nitrogen and alkalinity loadings. The results from this study demonstrate that together, these changes have only slightly altered the net advective flux of DIC into the Bay: the difference between DIC

river inputs and export to the coastal ocean has changed by only 6% over the past century (Figure 7a,b; Table 4). In contrast the changes in NEP and air-sea flux have been considerably larger. The Bay has become 19% more autotrophic over this time period (Figure 7a,b; Table 4), and the Bay's net air-sea flux has switched from being a small net source of $CO_2$ to the atmosphere, to a sink of $CO_2$ from the atmosphere. In the sections below, the causes of these overall changes, identified via the sensitivity experiments described above, are discussed individually, including both global changes (atmospheric $CO_2$ and

temperature) and regional watershed changes (riverine nitrogen, carbon and TA). In each case, there are mitigating factors that cause the changes to be lower than otherwise expected.

#### 4.2.1   Global changes and their impact on Chesapeake Bay carbonate chemistry

Between the early 1900's (1900–1914) and the early 2000's (2000–2014) atmospheric $CO_2$ concentration has increased by roughly 100 $\mu$atm. As expected, the impact of this single change on the inorganic carbon budget of the Chesapeake Bay

is significant, resulting in the transformation of the Bay from an average net source of $CO_2$ to the atmosphere (outgassing: $-20 \pm 80$ Gg-C yr$^{-1}$) to a net sink (ingassing: $+34 \pm 90$ Gg-C yr$^{-1}$). It is important to note that the standard deviations





associated with these interannual means in air-sea $CO_2$ flux represent interannual variability, and are significantly larger than the estimated long-term change. Thus, although the increase in atmospheric $CO_2$ is clearly increasing ingassing on average, there are still large year-to-year differences that may cause certain years in the early 1900's to be net sinks of atmospheric
$CO_2$ and certain years in the early 2000's to be net sources of atmospheric $CO_2$. This interannual variability makes it difficult to determine the average direction of the net air-sea $CO_2$ flux over the estuary unless long time series of climate-quality observations are available.

In addition to increases in atmospheric $CO_2$, atmospheric and estuarine temperatures have also been rising over the past century (Ding and Elmore, 2015; Muhling et al., 2018; Irby et al., 2018). The increased ingassing due to elevated atmospheric
$CO_2$ is partially mitigated (by roughly 50%; Figure 7c,d) via these increasing temperatures, which enhance $pCO_{2w}$ (because of solubility but also more respiration; Figure 7d). As a result, the change in air-sea $CO_2$ flux due to the global changes of the past century (+31 Gg-C yr$^{-1}$) is only ∼half as large as it would be without the concomitant increase in temperature. The increase in water temperature also leads to a 5% decrease in net ecosystem production through enhanced respiration of organic matter, consistent with heterotrophic processes being more sensitive to temperature than production (Lomas et al., 2002).

### 4.2.2 Regional watershed changes and their impacts on Chesapeake Bay carbonate chemistry

The increase in riverine DIN loading associated with urbanization and increased fertilizer usage has caused changes in the inorganic carbon budget over the last century that are nearly equal to those induced by global changes. Specifically, increased nitrogen loading has caused NEP to increase substantially (Figure 7e, +39%) and this, in turn, leads to lower surface DIC ($\sim -10$ mmol-C m$^{-3}$) and surface $pCO_{2w}$ ($\sim -25\,\mu$atm) in the southern half of the Bay. In response to these changes, the
net air-sea $CO_2$ flux into the Bay has increased considerably (Figure 7e, +70 Gg-C yr$^{-1}$), which is an even larger increase than that due to the higher atmospheric $CO_2$ (+54 Gg-C yr$^{-1}$). Another consequence of the enhanced NEP and lower DIC concentrations is a reduction in DIC export to the shelf (Figure 7e).

The historical increase in riverine carbon (TOC, DIC) and TA loadings has had relatively minor impacts on the inorganic carbon budget, as compared to those due to increased nitrogen loading discussed above (compare Figure 7e and f). Although
significantly increased DIC loading to the Bay is assumed (Table 2), and although the DIC concentrations of the Bay are increased substantially, much of the extra riverine DIC is simply exported to the coastal ocean (94%, Table 4). In terms of TOC, only 38% of the increase is exported to the coastal ocean. The remaining increase in TOC serves to increase respiration (decrease in NEP) partially offsetting the increase in production that resulted from the increased nitrogen loading discussed above. Regarding the air-sea $CO_2$ flux, the net effect of the increased respiration and increased riverine DIC/TA loadings is a
relatively small increase in $pCO_{2w}$ (approximately +6% on average over seasons and over the Bay between experiment $1900_C$ and the Control experiment) that brings the net air-sea $CO_2$ closer to being neutral. This ultimately serves to largely counteract the increased ingassing resulting from the increased nitrogen loading. Thus when considered together, the increases in nitrogen and carbon loading over the past century have resulted in the Chesapeake Bay becoming a greater sink for atmospheric $CO_2$ (by 27 Gg-C yr$^{-1}$), which is similar in magnitude to the increased sink due to global changes (+31 Gg-C yr$^{-1}$).





## 5 Summary and Concluding remarks

Sensitivity experiments were performed to isolate the effect of changes in: (1) atmospheric $CO_2$, (2) temperature, (3) riverine nitrogen loading and (4) riverine carbon and alkalinity loading, on the inorganic carbon balance of the Chesapeake Bay between the early 1900's and early 2000's. Both regional and global changes have enhanced the Bay's sink for atmospheric $CO_2$ by similar amounts. The increased riverine nitrogen load, a regional change, increased production which resulted in the Bay having a 19% higher (more autotrophic) NEP. Overall, the results of the study help clarify the impact that local management efforts (past or future) can have on the Bay's inorganic carbon balance and the limits of these efforts in the context of ongoing global changes. The temporal and spatial scope of this study also highlights the usefulness of modeling studies and how difficult it is to answer questions on these spatial and temporal scales from observations alone.

The comparison between the early 1900's and early 2000's suggests that the ongoing increase in atmospheric $CO_2$ concentrations overshadows the temperature-driven increase in $pCO_{2w}$ and outgassing. In other words, the Bay's trend toward more uptake of atmospheric $CO_2$ will likely continue in the decades to come. This is in contrast with regional changes in riverine loadings (mostly DIN and TA/DIC) which were particularly large in the past century and are not expected to continue in the future. Management efforts in the Bay's watershed, notably the implementation of a total maximum daily load (USEPA, 2010; Irby and Friedrichs, 2019), are expected to stabilize or reduce the nutrient inputs to the Bay over the next several decades. Similarly, Raymond and Oh (2009) (their Figure 1) suggest that coal production has stabilized in the past decades, and thus one would expect the Susquehanna's alkalinity and DIC concentrations to also stabilize. Overall, these results suggest that although changes in riverine nutrient inputs have played an important role in altering coastal carbon budgets over the past century, in the future ongoing global changes may have an even greater affect on coastal carbonate chemistry.

*Code and data availability.*  Documentation and source code for the numerical model used in this study (ROMS) are publically available at https://www.myroms.org/. Additional documentation about the biogeochemical equations and parameters is provided in the "Supplementary Material" of this publication. The model results used in the manuscript will be permanently archived on an online public repository (W&M ScholarWorks, https://scholarworks.wm.edu/) with a unique DOI after the manuscript is accepted.

*Author contributions.*  ES, MF and RN acquired the financial support and supervised the project leading to this publication. YY, HT and ES contributed model inputs and forcings necessary for the model simulations. MF and PS designed the numerical experiments and PS carried them out. PS analyzed the model results and created the figures+tables while all co-authors contributed to their interpretation. PS prepared the manuscript with contributions from all co-authors. PS was responsible for data curation of the model outputs.

*Competing interests.*  The authors declare that they have no conflict of interest.





*Acknowledgements.* This research was supported by the National Science Foundation (collaborative grants OCE-1537013, 1536996), NASA IDS (NNX14AF93G) and NASA Carbon Cycle Science to WETCARB (NNX14AM37G). This work used the Extreme Science and Engi-

neering Discovery Environment (XSEDE) supercomputer Comet at SDSC through allocation OCE-160013, which is supported by National Science Foundation grant number ACI-1548562. The authors also acknowledge William & Mary Research Computing (https://www.wm.edu/it/rc) for providing computational resources and/or technical support that have contributed to the results reported within this manuscript. This manuscript is contribution #xxxx of the Virginia Institute of Marine Science, William & Mary.





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



**Table 1.** List of experiments conducted in the study (see Section Methods for their description) with their differences from the Control experiment highlighted in bold

| Experiment | Atmos.$CO_2$ | Temperature | River N | River C |
|---|---|---|---|---|
| Ctrl.exp. | 2000–2014 | 2000–2014 | 2000–2014 | 2000–2014 |
| $1900_{CO2}$ | **1900–1914** | 2000–2014 | 2000–2014 | 2000–2014 |
| $1900_T$ | 2000–2014 | **1900–1914** | 2000–2014 | 2000–2014 |
| $1900_N$ | 2000–2014 | 2000–2014 | **1900–1914** | 2000–2014 |
| $1900_C$ | 2000–2014 | 2000–2014 | 2000–2014 | **1900–1914** |
| $1900_{all}$ | **1900–1914** | **1900–1914** | **1900–1914** | **1900–1914** |





**Table 2.** River concentrations of total alkalinity[a] (TA) and dissolved inorganic carbon[a] (DIC) for the 10 rivers of the model (see Section Methods)

| River | $\widehat{\text{TA}}$ | $\widehat{\text{DIC}}$ | $a_{\text{TA}}$ | $a_{\text{DIC}}$ |
| --- | --- | --- | --- | --- |
| | meq/m$^3$ | mmol/m$^3$ | meq/m$^3$ | mmol/m$^3$ |
| Susq. (2007) | 1050[b] | 1147[b] | 250 | 265 |
| Susq. (1900–1914) | 600 | 706 | 250 | 265 |
| Potomac | 1550 | 1680 | 375 | 405 |
| James | 1055 | 1180 | 300 | 335 |
| Patuxent | 975 | 975 | n/a | n/a |
| Rappahannock | 400 | 590 | n/a | n/a |
| York | 350 | 485 | n/a | n/a |
| Elk | 785 | 785 | n/a | n/a |
| Chester | 785 | 785 | n/a | n/a |
| Choptank | 785 | 785 | n/a | n/a |
| Nanticoke | 785 | 785 | n/a | n/a |

[a] Concentrations are parameterized as: $\text{DIC}(t) = \widehat{\text{DIC}} + a_{\text{DIC}} \cos\left(5\pi/4 - \omega t\right)$,
where $\omega = 2\pi/(365\,\text{days})$ and $t$ is days since year 0 (proleptic calendar).

[b] Value for year 2007 (the concentrations of the Susquehanna River include
a long-term trend during 2000–2014; see Section Methods).

n/a indicates that no seasonality is prescribed.





**Table 3.** Mean riverine loadings over the two periods of interest (see Section Methods)[a]

| Riverine loading | 1900–1914 | 2000–2014 |
| --- | --- | --- |
| Freshwater (km$^3$ yr$^{-1}$) | n/a[b] | 86 |
| DIN (Gg-N yr$^{-1}$) | 72 | 96 |
| TON (Gg-C yr$^{-1}$) | 46 | 47 |
| TA (Geq yr$^{-1}$) | 71 | 89 |
| DIC (Gg-C yr$^{-1}$) | 955 | 1169 |
| TOC (Gg-C yr$^{-1}$) | 457 | 507 |

[a] The values are for the 10 rivers of the model (combined).

[b] The experiments assume the same riverine freshwater discharge in both periods (see Methods).



**Table 4.** Carbon budget terms[a] (Gg-C yr$^{-1}$) for the Control experiment (2000–2014) and deviation[d] of the sensitivity experiments from the Control experiment (see Table 1 and Equations 5–7)

|  | Ctrl.exp. | Ctrl.exp. $-1900_{CO2}$ | Ctrl.exp. $-1900_T$ | Ctrl.exp. $-1900_N$ | Ctrl.exp. $-1900_C$ | Sum[c] | Ctrl.exp. $-1900_{all}$ |
|---|---|---|---|---|---|---|---|
| River DIC | $1169 \pm 350$[b] | 0 | 0 | 0 | +319 | +319 | +319 |
| Export DIC | $935 \pm 320$ | +51 | −13 | −31 | +300 | +308 | +305 |
| Riv−Exp.DIC | $234 \pm 80$ | −51 | +13 | +31 | + 19 | + 11 | + 14 |
| Air-sea flux | $34 \pm 90$ | +54 | −23 | +70 | −43 | + 58 | + 41 |
| $\partial DIC/\partial t$ | $9 \pm 70$ | + 2 | +2 | 0 | +5 | + 9 | + 5 |
| Production | $4748 \pm 360$ | 0 | +252 | +492 | +1 | +745 | +722 |
| Respiration | $4489 \pm 340$ | 0 | +265 | +391 | +30 | +685 | +673 |
| NEP | $259 \pm 60$ | 0 | −13 | +101 | −29 | + 59 | + 49 |
| River TOC | $507 \pm 140$ | 0 | 0 | 0 | +56 | + 56 | + 56 |
| Export TOC | $545 \pm 140$ | 0 | −4 | +46 | +22 | + 64 | + 63 |
| Riv−Exp.TOC | $-38 \pm 50$ | 0 | +4 | −46 | +34 | −8 | −7 |
| Burial | $221 \pm 20$ | 0 | −9 | +56 | +6 | + 52 | + 44 |
| $\partial TOC/\partial t$ | $-1 \pm 20$ | 0 | 0 | −1 | 0 | 0 | −1 |

[a] The values are averaged over the period of the simulation and rounded to the nearest integer.

[b] The symbol $\pm$ indicates the interannual variability (see Methods).

[c] "Sum" is the sum of the deviations associated with experiments $1900_{CO2}$, $1900_T$, $1900_N$ and $1900_C$.

[d] A version of this table with absolute values (rather than deviations) is available in the Supplementary Material.


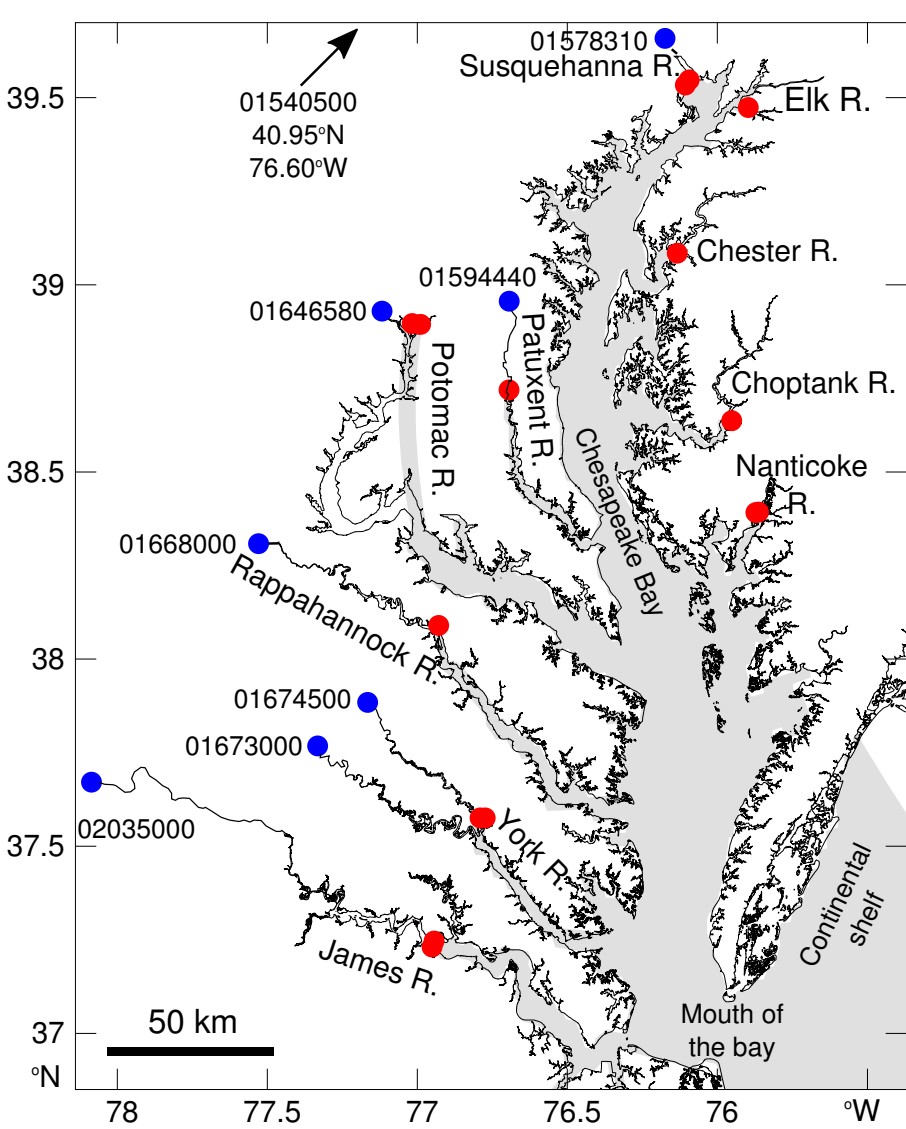

**Figure 1.** Map of the study area with the key tributaries labeled. The gray shading represents the model grid cells (see Da et al. (2018) for a map of the full model domain). Red circles represent the locations of riverine inflow in the model (10 rivers total). Blue circles represent locations where riverine alkalinity and DIC data are available. Each location is identified with an 8 digit number (see Section Methods).



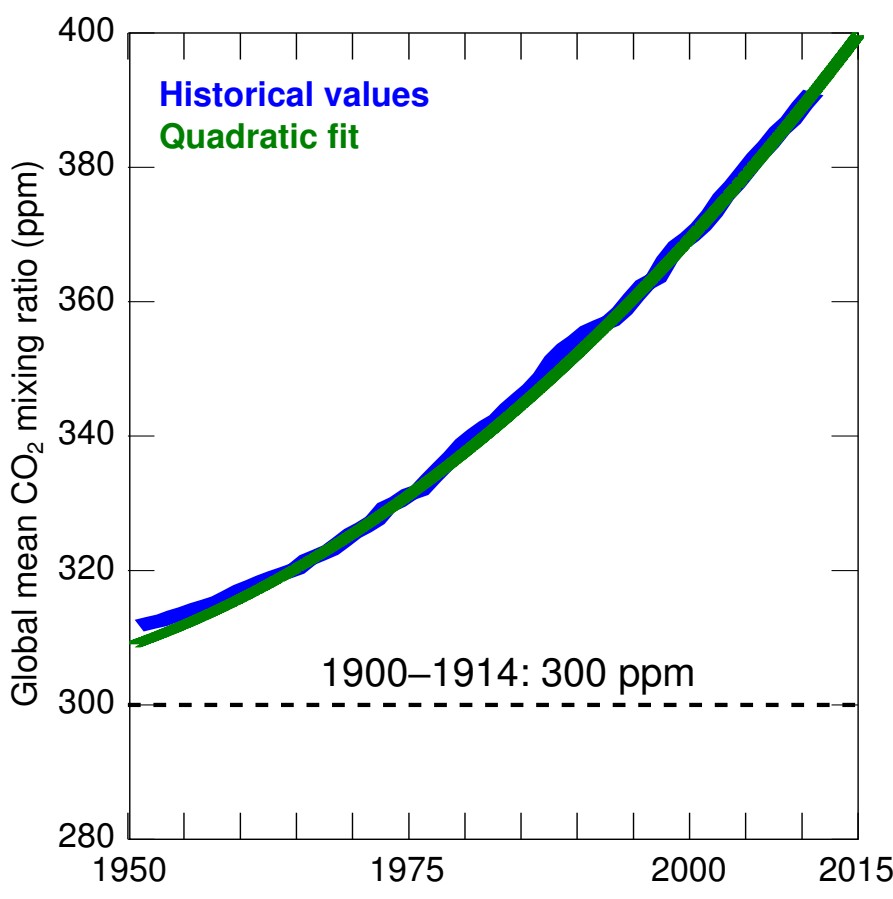

**Figure 2.** $CO_2$ mixing ratio used in the numerical experiments (green; see Section Methods). The historical values (blue) are from Miller et al. (2014). The dashed line represents the constant value assumed in the experiments of 1900–1914.

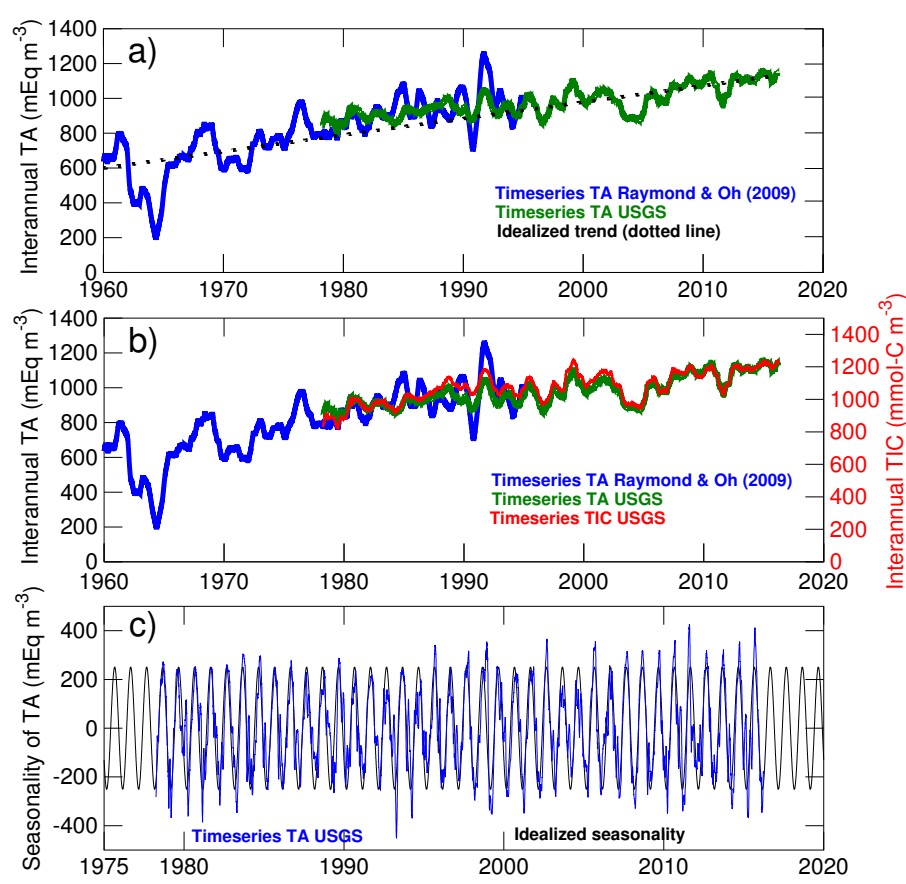

**Figure 3.** Long-term changes in the concentration of total alkalinity (TA) and dissolved inorganic carbon (DIC) in the Susquehanna River. (a) Comparison between TA timeseries from two locations in the river (see Section Methods). The interannual variability of the timeseries is emphasized with a 1 year moving average. The dotted line represents the idealized linear trend used in the Control experiment. (b) Comparison between timeseries of TA and DIC. (c) Comparison between the seasonality of TA and the idealized seasonal cycle used in model simulations.



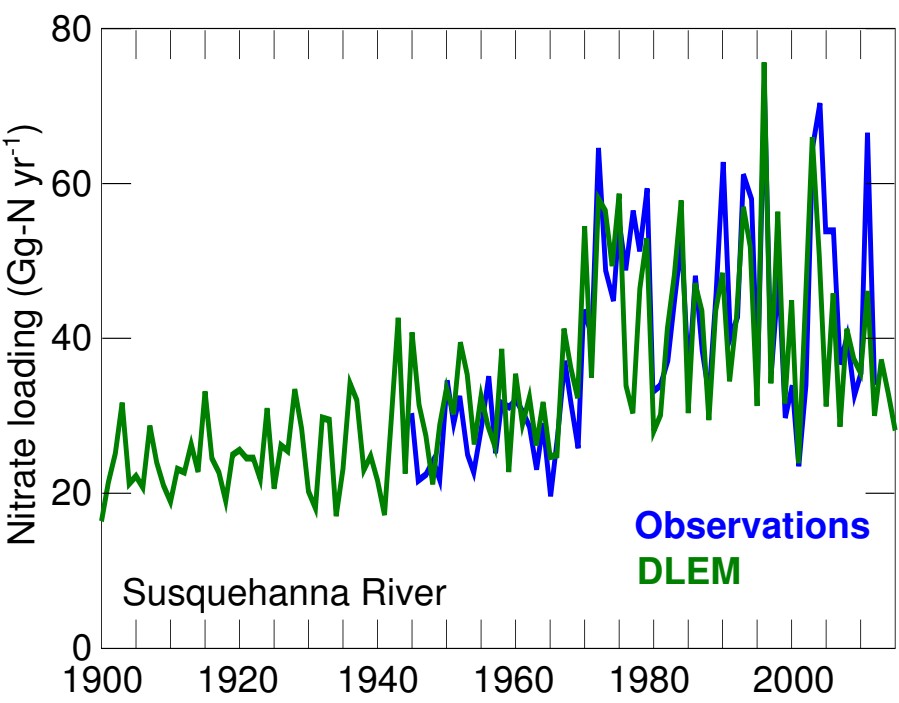

**Figure 4.** Long-term changes in the nitrate loading of the Susquehanna River. Observations are from Harding et al. (2016) (their Figure 7) while model values are from DLEM (see Section Methods).



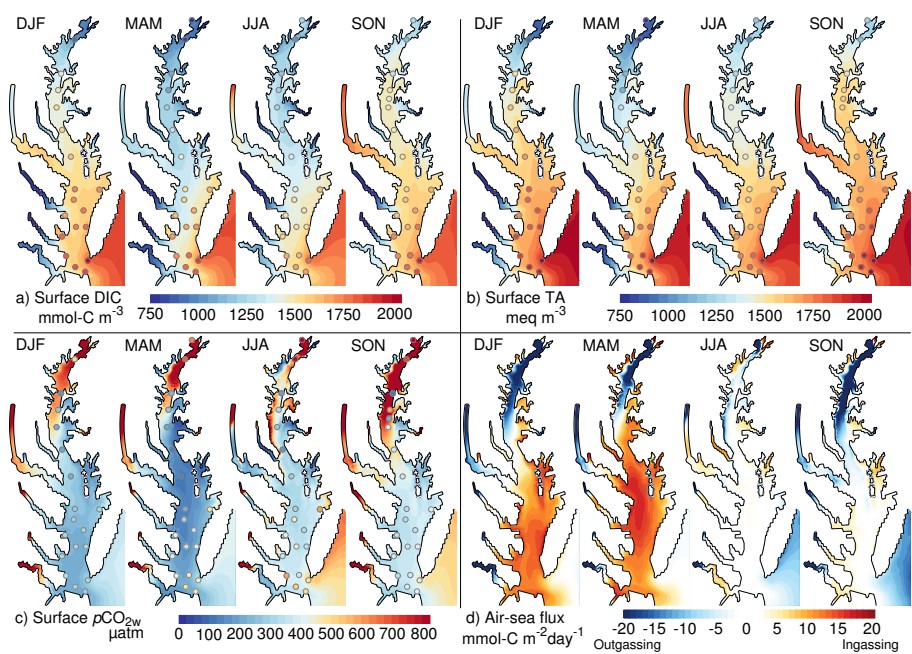

**Figure 5.** Overview of the spatial and seasonal variability of the inorganic carbon system. The shading represents a seasonal climatology from the Control experiment (years 2000–2014). The circles are derived from observations (see Section Methods). DJF is Dec. to Feb., MAM is March to May, JJA is June to Aug., and SON is Sep. to Nov.





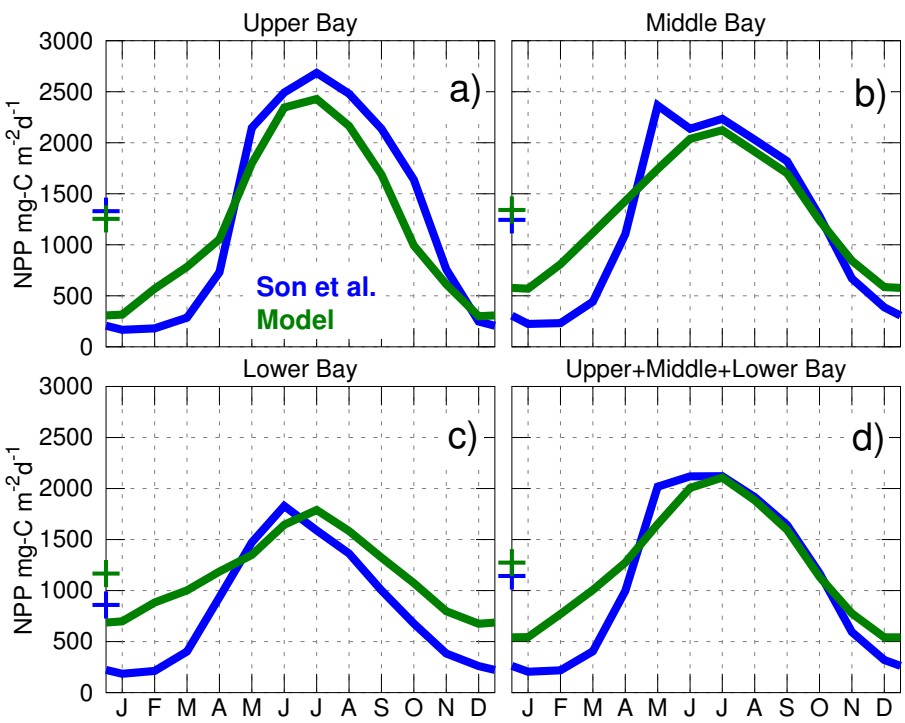

**Figure 6.** Seasonal climatology of the net primary production (NPP) over the period 2002–2011. The blue curves are the modeled results and the green curves are from the empirical model of Son et al. (2014) (their Figure 7). The upper, middle and lower Bay regions are defined as in Son et al. (2014) (their Figure 1). The '+' symbols represent the annual mean value of the curves. The model results are from the Control experiment (Table 1).

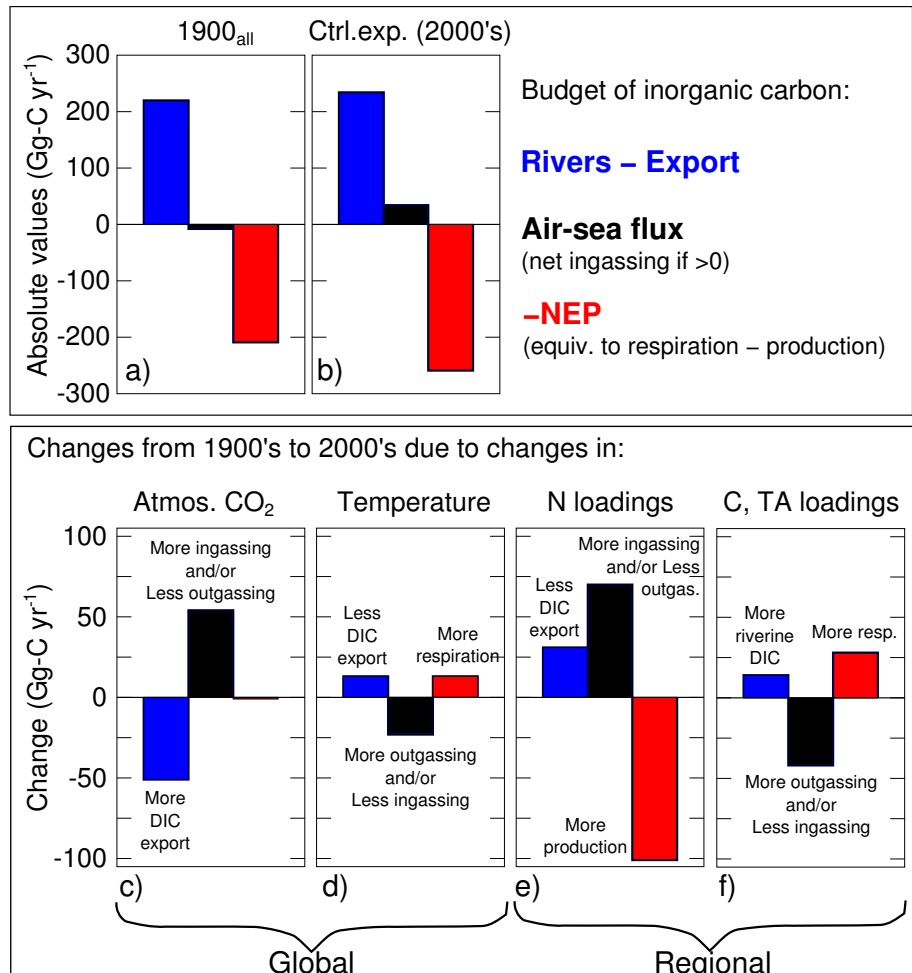

**Figure 7.** Summary of the changes in the inorganic carbon system for the six model experiments (Table 1). "Rivers minus Export" combines the riverine DIC input and the export of DIC at the Bay's mouth (Table 4).