# Peer review of "Relative impacts of global changes and regional watershed changes on the inorganic carbon balance of the Chesapeake Bay"

_Biogeosciences, 2020_

## Referee Comment (RC1) · Anonymous Referee #1 · 25 Apr 2020

General comments: The authors use a linked land-estuarine-ocean model to explore the inorganic carbon balance in Chesapeake Bay. Several sensitivity scenarios are conducted to determine the relative impacts of global changes and regional watershed changes on the inorganic carbon budget. These scenarios include a control experiment with realistic forcing of a period of 15 years from 2000 to 2014, an air pCO2 change experiment, a temperature change experiment, a riverine nutrient change experiment, a carbon and alkalinity change experiment and a combined change experiment to represent the period of 1900-1914.

The carbonate system was validated by comparing model outputs against a variety of

field observations along the main channel. The model displayed strong spatiotemporal patterns of DIC, Alkalinity, pCO2. This study successfully quantified the contributions of variable anthropogenic stressors on the inorganic carbon balance. The global pCO2 increase has enhanced bay-wide in-gassing, which, however, is mitigated by the temperature increase. Regional nutrient loading increase can enhance the in-gassing by increasing the NEP. Differently, the riverine carbon and alkalinity increase would reduce the in-gassing process. The manuscript is very well written, clear, and should be published with some minor revisions.

Specific comments: Line 118-119. Due to limited observations of DIC and TA, the author use the salinity derived DIC and TA as the forcing at the ocean side. It would be helpful to mention the pH range calculated with these salinity derived DIC/TA, making sure the pH is in a reasonable range.

Line 124. The 50 anthropogenic DIC might represent a small change to surface/bottom DIC, however, this DIC change could affect the surface water pCO2 a lot and have a much larger impact on the air-sea gas exchange.

Line 146. Why not use the calculated DIC (from pH and the TA you prepared), which could be more accurate to represent the riverine forcing?

Section 3.1.1 Please provide some quantitative measures (e.g. RMSE, relative error) either in Figure 5 or in the texts. It's hard to see the performance of the model in carbonate system.

---

## Referee Comment (RC2) · Anonymous Referee #2 · 27 Apr 2020

The manuscript describes a model sensitivity experiment assessing the impacts of atmospheric carbon dioxide, water temperature and riverine nitrogen, carbon and alkalinity on the inorganic carbon budget of Chesapeake Bay. Model sensitivity experiments are an essential tool for understanding the individual and combined impacts of different components of complex non-linear systems. The experiment is well designed. The modelling system used is based on an established published model; modifications for the current work are clearly described and model validation is included. The result that the two global changes (temperature and CO2 concentrations) have opposite impacts on air-sea CO2 flux is expected, but the experiments also show mitigating impacts on DIC export and Net Ecosystem Production and give estimates of all magnitudes. Like-

wise, the impacts of two regional changes (riverine nitrogen and carbon loads) partially mitigate changes in air-sea CO2 fluxes and NEP. The results are interesting and give an insight into the likely future carbon budget in Chesapeake Bay. The manuscript is well written and structured, with appropriate figures and tables.

In the sensitivity experiments, the meteorological forcing is the same as the control simulation (early 2000s) and the water temperature and riverine DIC and alkalinity experiments use values estimated from mid 20th century data. To avoid any confusion for the reader, it would be useful to reinforce (perhaps in the conclusion section) that the sensitivity experiments are not modelling actual early 1900s conditions.

One technical correction: the labels of figure 3b refer to TIC; DIC is used in the caption.

---

## Author Response (AR1)

**Response to referees' comments, Manuscript bg-2020-117**

St-Laurent, P., M.A.M. Friedrichs, R.G. Najjar,
E.H. Shadwick, H. Tian, Y. Yao, E.G. Stets

June 13, 2020

**Cover letter to the Editor**

Dear Editor of Biogeosciences,

Please find enclosed a revised copy of the manuscript "Relative impacts of global changes and regional watershed changes on the inorganic carbon balance of the Chesapeake Bay" by P.St-Laurent, M.Friedrichs, R.Najjar, E.Shadwick, H.Tian, Y.Yao and E.Stets. It includes the revised manuscript, a point-by-point reply to the comments (this document), and a marked-up manuscript version showing the changes made.

The changes made to the manuscript directly follow the suggestions of the referees (as detailed on the Discussion board of the journal; we repeated the referees' comments (and our response to them) below for completeness). We made additional minor changes to satisfy a federal review process from the U.S. Geological Survey (the employer of coauthor E.Stets). All the changes made to the manuscript are clearly highlighted in the marked-up manuscript version.

The authors have no conflict of interest to report and they agree with this submission. Thank you for your consideration.

Pierre St-Laurent (corresponding author)

**Authors' response to comments from Anonymous Referee #1**

We thank the referee for their careful reading of the manuscript and for providing helpful and thoughtful comments. Referees' comments are italicized while the Authors' responses are not italicized.

*General comments: The authors use a linked land-estuarine-ocean model to explore the inorganic carbon balance in Chesapeake Bay. Several sensitivity scenarios are conducted to determine the relative impacts of global changes and regional watershed changes on the inorganic carbon budget. These scenarios include a control experiment with realistic forcing of a period of 15 years from 2000 to 2014, an air pCO2 change experiment, a temperature change experiment, a riverine nutrient change experiment, a carbon and alkalinity change experiment and a combined change experiment to represent the period of 1900-1914.*
*The carbonate system was validated by comparing model outputs against a variety of field observations along the main channel. The model displayed strong spatiotemporal patterns of DIC, Alkalinity, pCO2. This study successfully quantified the contributions of variable anthropogenic stressors on the inorganic carbon balance. The global pCO2 increase has enhanced bay-wide in-gassing, which, however, is mitigated by the temperature increase. Regional nutrient loading increase can enhance*

*the in-gassing by increasing the NEP. Differently, the riverine carbon and alkalinity increase would reduce the in-gassing process. The manuscript is very well written, clear, and should be published with some minor revisions.*

*Specific comments: Line 118-119. Due to limited observations of DIC and TA, the author use the salinity derived DIC and TA as the forcing at the ocean side. It would be helpful to mention the pH range calculated with these salinity derived DIC/TA, making sure the pH is in a reasonable range.*

We agree with the reviewer that this is a valuable sanity check. We added a new sentence to the manuscript that includes this information:

"...are combined with the seasonal climatology used for salinity to prescribe TA and DIC at the model open boundary. The pH at the oceanic model boundary calculated from these TA and DIC values varies seasonally and spatially within the range $7.75 < \text{pH} < 8.05$ with an average value $\text{pH} = 7.89$ (total scale). This range is consistent with the measurements in Wang et al. 2013 (their Figure 8b, transect "MA", $\text{pH} \approx 7.9 \pm 0.1$ where $\pm$ represents one standard deviation). Note that the same oceanic conditions..."

Reference cited: Wang, Z.A., R. Wanninkhof, W.J. Cai, R.H. Byrne, X. Hu, T.-H. Peng, W.J. Huang, 2013, The marine inorganic carbon system along the Gulf of Mexico and Atlantic coasts of the United States: Insights from a transregional coastal carbon study, Limnol. Oceanogr., 58(1), 325-342, https://doi.org/10.4319/lo.2013.58.1.0325

*Line 124. The 50 anthropogenic DIC might represent a small change to surface/bottom DIC, however, this DIC change could affect the surface water pCO2 a lot and have a much larger impact on the air-sea gas exchange.*

We agree with the referee that the original sentence wasn't properly acknowledging the potential impact of anthropogenic DIC on the continental shelf. The referee's comment convinced us that we shouldn't speculate on this matter, and that we should simply state in the text that this component should be considered in future studies. We thus replaced the original passage by:

"...the same oceanic conditions are used in the 1900-1914 and 2000-2014 experiments since we are primarily interested in historical changes that occurred inside the Bay and at its surface (*e.g.*, atmospheric $CO_2$). The potential impact of the historical change in DIC on the continental shelf (i.e., the anthropogenic DIC) is thus not represented here, but it should be considered in future studies."

*Line 146. Why not use the calculated DIC (from pH and the TA you prepared), which could be more accurate to represent the riverine forcing?*

We appreciate the referee's question. The calculated DIC (shown in Figure 3b of the manuscript) exhibits variability on multiple timescales that most likely reflects different processes affecting DIC in rivers. In the context of the present manuscript, we are specifically focusing on the *long-term change* in DIC (which we parameterize as a linear increasing trend). The remaining interannual variations are considered beyond the scope of the present study. We included an additional sentence in the manuscript to clarify this point:

"...TA and DIC followed similar trajectories over these decades (Figure 3b); therefore we also assume linearly increasing DIC in the 2000-2014 experiment (10 mmol-C/m3/yr). The remaining year-to-year variability apparent in Figure 3b is considered beyond the scope of the study and not represented in the model experiments. Finally, a seasonal cycle in TA and DIC..."

*Section 3.1.1 Please provide some quantitative measures (e.g. RMSE, relative error) either in Figure 5 or in the texts. It's hard to see the performance of the model in carbonate system.*

(N.B. In our response, we assume the referee is referring to Section 3.1.2 ("Evaluation of the modeled inorganic carbon system"), and not Section 3.1.1.)

We followed the referee's advice and computed quantitative measures of the model skill for the carbonate system. We integrated these values in the existing text of Section 3.1.2 during the revision process (please see the "track-change" version of the manuscript).

**Authors' response to comments from Anonymous Referee #2**

We thank the referee for their careful reading of the manuscript and for providing helpful and thoughtful comments. Referees' comments are italicized while the Authors' responses are not italicized.

*The manuscript describes a model sensitivity experiment assessing the impacts of atmospheric carbon dioxide, water temperature and riverine nitrogen, carbon and alkalinity on the inorganic carbon budget of Chesapeake Bay. Model sensitivity experiments are an essential tool for understanding the individual and combined impacts of different components of complex non-linear systems. The experiment is well designed. The modelling system used is based on an established published model; modifications for the current work are clearly described and model validation is included. The result that the two global changes (temperature and CO2 concentrations) have opposite impacts on air-sea CO2 flux is expected, but the experiments also show mitigating impacts on DIC export and Net Ecosystem Production and give estimates of all magnitudes. Likewise, the impacts of two regional changes (riverine nitrogen and carbon loads) partially mitigate changes in air-sea CO2 fluxes and NEP. The results are interesting and give an insight into the likely future carbon budget in Chesapeake Bay. The manuscript is well written and structured, with appropriate figures and tables.*
*In the sensitivity experiments, the meteorological forcing is the same as the control simulation (early 2000s) and the water temperature and riverine DIC and alkalinity experiments use values estimated from mid 20th century data. To avoid any confusion for the reader, it would be useful to reinforce (perhaps in the conclusion section) that the sensitivity experiments are not modelling actual early 1900s conditions.*

The referee makes a very good point. We inserted the following statement in Section "Summary and Concluding remarks":

"...experiments were performed to isolate the effect of changes in: (1) atmospheric CO2 , (2) temperature, (3) riverine nitrogen loading and (4) riverine carbon and alkalinity loading, on the inorganic carbon balance of the Chesapeake Bay between the early 1900's and early 2000's. Limited information is available for the early 1900's and thus these experiments are meant to highlight the aforementioned changes rather than to model actual early 1900's conditions. Both regional and

global changes..."

*One technical correction: the labels of figure 3b refer to TIC; DIC is used in the caption.*

We apologize for this oversight. The figure has been corrected accordingly during the revision of the manuscript.

[revised manuscript text omitted]